# Interferon-Gamma Secretion Is Significantly Decreased in Stage III Breast Cancer Patients

**DOI:** 10.3390/ijms25084561

**Published:** 2024-04-22

**Authors:** Jung Im Yi, Jean Schneider, Seung Taek Lim, Byeongkwan Park, Young Jin Suh

**Affiliations:** 1Department of Surgery, The Catholic University of Korea St. Vincent’s Hospital, Suwon 16247, Republic of Korea; jenhome1210@gmail.com (J.I.Y.); in-somnia@hanmail.net (S.T.L.); bkarchfiend.bkp@gmail.com (B.P.); 2School of Medicine, Texas Tech University Health Sciences Center, Lubbock, TX 79430, USA

**Keywords:** breast cancer, natural killer cells, NK cells, interferon-gamma

## Abstract

Even though some studies have shown possible clinical relationship between molecular subtypes and tumor infiltrating natural killer (NK) cells around tumors, there are few studies showing the clinical relevance of peripheral NK cell activity at diagnosis in female patients with invasive breast cancer. A total of 396 female invasive breast cancer patients who received curative surgical treatment from March 2017 to July 2021 were retrospectively analyzed. NK cell activation-induced interferon-gamma (IFN-γ) secretion measured by enzyme-linked immunosorbent assay was used to measure the activity of peripheral NK cells. Statistical analyses were performed to determine clinical relationships with major clinicopathologic parameters. Quadripartite NK cell activity measured by induced interferon-gamma showed significant relevance with staging and body mass index, and some of the inflammatory serological markers, namely N/L (neutrophil/lymphocyte), P/N (platelet/neutrophil), and P/L (platelet/lymphocyte), showed significantly different NK activity in each interval by univariate analysis. A binary subgroup analysis, setting the IFN-γ secretion cut-off at 100 pg/mL, showed that stage III was significantly increased and axillary lymph node metastasis positivity was increased in the group of IFN-γ < 100 pg/mL, and IFN-γ secretion decreased with an increasing N stage, increased BMI (body mass index), and decreased production of IFN-γ. Following this, the same binary analysis, but with the IFN-γ secretion cut-off at 250 pg/mL, also showed that secretion in stage III was increased in those concentrations with <250 pg/mL, axillary lymph node positivity appeared to be correlated, and BMI ≥ 30 increased in prevalence. Additional ANOVA post hoc tests (Bonferroni) were performed on some serological markers that tended to be somewhat inconsistent. By subgroup analysis with Bonferroni adjustment between the IFN-γ secretion and TNM stage, no significant difference in IFN-γ secretion could be identified at stages I, II, and IV, but at stage III, the IFN-γ secretion < 100 pg/mL was significantly higher than 250 ≤ IFN-γ secretion < 500 pg/mL or IFN-γ secretion ≥ 500 pg/mL. According to this study, stage III was significantly associated with the lowest IFN-γ secretion. Compared to a higher level of IFN-γ secretion, a lower level of IFN-γ secretion seemed to be associated with increased body mass index. Unlike when IFN-γ secretion was analyzed in quartiles, as the IFN-γ secretion fell below 100 pg/mL, the correlation between axillary lymph node positivity and increased N stage, increased BMI, and increased N/L and P/L, which are suggested poor prognostic factors, became more pronounced. We think a peripheral IFN-γ secretion test might be convenient and useful tool for pretreatment risk assessment and selecting probable candidates for further treatment such as immunotherapy in some curable but high-risk invasive breast cancer patients, compared to other costly assaying of tissue NK cell activity at diagnosis.

## 1. Introduction

Globally, breast cancer is the most common cancer threatening women’s health, and over the past two decades, improvements in diagnostics and the development of taxanes and targeted agents has led to significant improvements in treatment, with a positive impact on survival rates. However, for triple-negative breast cancer, which has the poorest prognosis due to its lack of hormone receptors and poor response to targeted agents, recent clinical studies are slowly raising hopes for the potential of immunotherapy with atezolizumab or pembrolizumab in addition to conventional treatment. Host immunity is crucial for the oncogenesis from initiation to metastatic progression [1]. The host immune system has the potential not only for the specific destruction of tumor cells without noxious stimuli, but also to promote tumor growth through a process called immunoediting. Immunoediting comprises three phases: elimination, equilibrium, and escape [2]. Elimination is achieved through the identification and destruction of the more immunogenic cancer cells by cytotoxic immune cells, characterized by the infiltration of effector cells of the innate and adaptive immune system [2,3]. However, malignant progression is accompanied by profound immune suppression that interferes with an effective anti-tumor response and tumor elimination in many immunogenic cancers [2,3]. In particular, as the first line of defense, the innate immune cells including macrophages, neutrophils, dendritic cells, interferon-producing cells, natural killer (NK) cells, and innate lymphoid cells are initially involved in incipient tumor formation and facilitate cellular transformation and malignant development [2,4,5]. Among these cells, NK cells are an important subset of innate lymphoid cells. In contrast to tumor-associated macrophages and neutrophils, which can exert either pro- or anti-tumor roles in tumor progression, NK cells are devoted anti-tumor contenders [2]. The importance of NK cells in controlling tumor growth by interacting directly with tumor cells or affecting the function of other cellular components of innate and adaptive immunity in the tumor microenvironment has been demonstrated in different experimental mouse cancer models [6]. In clinical practices, the presence of tumor-infiltrating lymphocytes within primary cancer lesions, including CD8+ cytotoxic T lymphocytes and peritumor-infiltrating NK cells, has been known to be associated with neoadjuvant chemotherapeutic efficacy in terms of the post-treatment reduction in tumor size and with prolonged disease-free survival of breast cancer patients [7,8,9,10]. However, the precise prognostic and predictive role of peripheral blood NK cell activity in human breast cancer patients at diagnosis remains to be further evaluated, especially in the era of immunotherapy, where breast cancer is one of the solid tumors that may be treated with immunotherapeutic agents in some patients. Although previous studies have shown significantly depressed NK cell activity in patients with breast cancer along with progression [11,12], and have shown a correlation between the systemic activation of peripheral blood NK cells after neoadjuvant chemotherapy and the disappearance of axillary lymph node metastasis [13,14], this aspect is very vaguely documented in the literature and not supported with real values. Additionally, we have seen contradictory clinical outcomes on the role of tumor-infiltrating lymphocytes from patients showing aggressive clinical phenotype while having indolent clinicopathologic factors or vice versa. Although several immunotherapeutic agents are beginning to show promise in some breast cancer phenotypes to improve treatment outcomes when combined with conventional therapy, breast cancer is not a highly immunogenic cancer, as we know. Therefore, it is important to select the right patients in order to achieve more effective immunotherapeutic outcomes in breast cancer. In this retrospective study, we investigated how a quantitative ELISA essay of NK cell activation-induced interferon-gamma (IFN-γ) secretion correlates with various proven clinicopathological factors of breast cancer at the time of diagnosis, as an alternative to the complex and time-consuming immunohistological evaluation of the role of tumor-infiltrating lymphocytes, which has previously shown conflicting results.

## 2. Results

All 396 patients who received curative surgical treatment for invasive breast cancer were analyzed (The Table in Section 4). All measured values of concentration of interferon-gamma were precisely categorized into one of the following four ranges (pg/mL): <100, 100~249, 250~499, 500, or more. The median age for all ranges was not different. The menstrual status composition was not different either. Neither T stage nor N stage classified as either axillary node positivity or N0 to N3 was statistically significant. The tumor size measured by maximum dimension (cm) was irrelevant across the four categories. However, tumor staging was significantly associated with decreased higher IFN-γ concentration (*p* = 0.001). Body surface area was not associated with IFN-γ concentration, but there was a significant trend toward decreased IFN-γ secretion with an increasing body mass index (*p* = 0.021). The lowest and highest quartiles of BMI were difficult to interpret due to the small number of patients. The index tumor location, molecular subtype and Ki-67 index did not show significant differences. Among the inflammatory serological markers, N/L, P/N, and P/L showed differential IFN-γ concentration in each interval (*p* = 0.02, 0.021, and 0.001, respectively). We analyzed IFN-γ concentration by dividing it into quartiles as recommended by the assay kit, but given the lack of a sufficient rationale for how to set the boundaries of each quartile, we re-analyzed the data by dividing the IFN-γ concentration levels into two groups with boundaries of 100, 250, and 500 pg/mL, respectively (Table 1).

Then, we set the IFN-γ concentration cut-off at 100 pg/mL and analyzed it, and found that stage III was significantly increased (*p* < 0.001) and axillary lymph node metastasis positivity was increased (*p* = 0.035) under 100 pg/mL, and IFN-γ concentration decreased with an increasing N stage (*p* = 0. 007), increased BMI, and decreased production of IFN-γ (*p* = 0.033), and we found that N/L and P/L values were inversely correlated with a decreased production of IFN-γ (*p* = 0.016 and 0.006, respectively) (Table 1a). Unlike when the IFN-γ concentration was analyzed in quartiles, as the IFN-γ concentration fell below 100 pg/mL, the correlation between axillary lymph node positivity and increased N stage, increased BMI, and increased N/L and P/L became more pronounced.

Next, we set the IFN-γ concentration cut-off at 250 pg/mL and performed the same analysis by splitting the total in two, and we found that secretion in stage III increased in those concentrations with <250 pg/mL (*p* = 0.007), and axillary lymph node positivity appeared to be correlated (*p* = 0.047), but when we subdivided the N stage, the difference disappeared, BMI ≥ 30 increased in prevalence (*p* = 0.038), and there was a significant increase in N/E in addition to N/L and P/L (*p* = 0.005, 0.001, and 0.025, respectively).

Finally, we set the IFN-γ concentration cut-off to 500 pg/mL and analyzed the two groups, which showed a similar trend to the 250 pg/mL cut-off. However, the analysis based on BMI did not show a significant difference, and the other factors followed the same pattern.

Additional ANOVA post hoc tests (Bonferroni) were performed on some serological markers that tended to be somewhat inconsistent: N/L increased more when the IFN-γ concentration was <100 pg/mL than when it was ≥500 pg/mL (*p* = 0. 017), P/N increased more when the IFN-γ concentration was <100 pg/mL than when it was 250~500 pg/mL (*p* = 0.031), and P/L increased significantly when the IFN-γ concentration was <100 pg/mL than when it was 250–500 pg/mL and ≥500 pg/mL (*p* = 0.025 and 0.001, respectively) (Table 2).

When we perform a subgroup analysis by applying a Bonferroni adjustment between the IFN-γ concentration and TNM stage (Table 3), the threshold for a statistically significant *p*-value in this analysis is 0.0083, which is one-sixth of 0.05. Resetting the new significance criterion as of *p* = 0.0083, no significant difference in the IFN-γ concentration could be identified at stages I, II, and IV, but at stage III, the IFN-γ concentration < 100 pg/mL was significantly higher than 250 ≤ IFN-γ concentration < 500 pg/mL and IFN-γ concentration ≥ 500 pg/mL (*p* = 0.004 and <0.001, respectively).

Body surface area and body mass index (BMI: kg/m^2^) failed to show a significant difference among four ranges of interferon-gamma level. However, the proportion of BMI values between 25 and 30 was significantly higher in patients with interferon-gamma less than 100 pg/mL, and the percentage of patients with a BMI of 30 or more was exceptionally higher in interferon-gamma between 100 and 250 pg/mL (*p* = 0.021).

Several analyses of the correlation between body mass index and IFN-γ concentration have shown that the results are somewhat inconsistent across criteria. Therefore, we also performed subgroup analyses using Bonferroni adjustment (Table 4). As in the TNM staging analysis, the criterion for significance is *p* = 0.0083, 1/6 the value of 0.05. We rechecked for significant differences using this new criterion. We divided BMI into quartiles and set cut-off values of 18.5, 25, and 30 kg/m^2^ for each. As a result, we checked the difference in IFN-γ concentration for each BMI group and found no significant difference. Considering the impact of the number of patients in the analysis, we analyzed two more cases by dividing the BMI bands into tertiles, but no significant differences in IFN-γ concentration were found.

Index tumor location failed to show significance according to interferon-gamma level. Interestingly, the molecular subtype had no relationship with increased or decreased levels of interferon-gamma. There was no significant result associating Ki-67 levels at diagnosis with IFN-γ concentration.

Using log rank (Mantel–Cox) analysis, we determined the significance between the TNM stage and IFN-γ concentration (Figure 1). Due to the small number of patients with stage IV disease, it was not possible to obtain the analysis using quartile IFN-γ concentration cut-off and after setting the cut-off at 500 pg/mL. Other than that, the analyses were possible, but we failed to show a statistically significant relationship between survival and IFN-γ concentration.

## 3. Discussion

In this study, we assessed the correlation between the pretreatment peripheral blood NK cell activity measured by released interferon-gamma and the clinicopathological characteristics of patients diagnosed with curable invasive breast cancer. Natural killer cell activity has been shown to have a possible role in improving clinical outcomes by immune activation in the tumor microenvironment [14]. In that study, despite showing clinical data from 39 patients, the authors used the ^52^Cr-labeling method to measure peripheral NK cell activity and presented a possible role of systematically activated NK cells in improving cancer cell elimination. In our study, we avoided using radioisotopes to measure NK cell activity by using an ELISA kit. In addition, an NK cell activity test showed its possible application for risk assessment and early diagnosis in other solid cancers [15]. In breast cancer, NK cells have been shown to have a probable positive role in triple-negative breast cancer as significantly aggregated infiltrates close to the tumor which may translate into a good prognosis after proper clinical use and thus overcome heterogeneity in these patients [16]. Anti-HER2 monoclonal antibody trastuzumab stimulates breast cancer cells and NK cells to produce TGF-β and interferon-γ, respectively [17]. Consequently, PD1 expression on NK cells is induced by TGF-β, which in turn results in enhanced NK cell cytotoxicity by PD1 blockade as well. This can be effective way to fight against trastuzumab-resistance in combination with HLA-G blockade. Although this kind of accumulating evidence might predict some possible and promising relationships with peripheral NK cell activity measured by induced IFN-γ, we failed to obtain positive relationships with these subtypes. Nonetheless, the immaturity of NK cells may be a promoter to progress triple-negative breast cancer that mandates further larger evaluation to determine the definitive role of mature and immature NK cells on the prognosis of this subtype [18]. In that study with a murine model, triple-negative breast cancer-associated immature NK cells increased poor clinical outcome. Recently, significantly impaired NK cells presented as a decreased absolute count can be used to evaluate a breast cancer patient’s immune status by susceptible biomarker [19].

We have shown that IFN-γ < 100 pg/mL in stage III was significantly different from the rest of the bands (100 ≤ IFN-γ ≤ 250, 250 ≤ IFN-γ ≤ 500, and IFN-γ > 500 pg/mL). The possible association of bands with stages, as shown in the Table in Section 4, was further analyzed by dividing the IFN-γ concentration into two bands, instead of using the previous cut-off values of 100, 250, and 500 pg/mL (Table 1), and was finally confirmed by Bonferroni adjustment in Table 3. Initially, stages I, II, and IV were also analyzed altogether, and a statistically significant difference was found only in stage III. In theory, in breast cancer, like other solid tumors, the more fragile the host immune system, the more likely it is that the cancer will leave the breast and spread to the axillary lymph nodes and lead to systemic metastases, or develop into locally advanced disease. However, this is not always the case, which makes it difficult to interpret the results of this study. In other words, although the analysis using IFN-γ concentration divided into four quartiles, based on the results in people without cancer, confirmed the expected finding that the IFN-γ concentration tends to decrease with increasing stage, excluding non-invasive stages 0 and IV of the worst prognosis. However, the fact that we were unable to prove meaningful results with respect to T (tumor size) and N (nodal involvement) along with TNM staging confirms that a new cut-off value is needed for in-depth analyses. The results of the binary analyses, which showed significant correlations with factors known to be associated with poor prognosis and which are close to the expected results, even more so than the four-arm analysis, suggest that peripheral IFN-γ concentration may have the potential to further increase the predictive value when used in conjunction with conventional TNM staging. Given that many patients with locally advanced hormone-dependent subtypes, not to mention triple-negative subtypes who are more likely to be candidates for immunotherapeutic agents with a promising oncologic outcome, may be limited by conventional therapy, further studies are warranted in these patients, and changes in NK cell activity associated with primary or secondary resistance in HER2-positive or luminal B subtypes should be explored beyond this study. While this study did not show a valid relationship for these matters, it is certainly the subjects worthy of further analyses with lots of data. Although the Ki-67 proliferation index, unlike other proven prognostic factors, is used in some criteria for defining molecular subtypes, it has not yet been established as a prognostic predictor, but given that it is an index of cancer growth rate, it is not unreasonable to assume that it has some interaction with host immunity in addition to correlation with the microenvironment surrounding breast cancer cells. However, this study did not show any significant results with IFN-γ concentration and Ki-67. This may require further analysis with different cut-off values for each subtype to understand the exact correlation. Taken together, IFN-γ concentration cut-off values of 100 or 250 pg/mL were associated with significantly higher stage, axillary node positivity and BMI as IFN-γ concentration decreased, with the exception of BMI, which was the same at 500 pg/mL. These results provide indirect evidence of a distinct relationship between these factors and patients’ NK cell activity. This is in contrast to the many conflicting results previously reported by researchers on the relationship between breast cancer and tumor-infiltrating lymphocytes (TILs), in which many researchers predicted would show theoretically consistent results. Considering the fact breast cancer is a systemic disease, not a localized disease, if the current direction of research is more narrowly focused on the true immune system of breast cancer patients, it can be hypothesized that the assessment of NK cell activity, which is consistent in fighting cancer among the various immune cells involved in the first defense against cancer, may be inaccurate based on TIL analysis of surrounding tissues of cancer. Rather, serial quantitative measurements of peripheral NK cells may provide a more objective, accurate, and timely cross-section of the innate immunity of breast cancer patients at each stage. It would have been nice to be able to see the same statistical significance in stage IV as in stage III, but given that there were situations where the number of patients in stage IV at diagnosis was so small that the statistics were problematic, we would expect to see similar results if we could obtain a sufficient number of stage IV patients to facilitate a statistical analysis. In terms of BMI, the increment of BMI seems to be related to the increased risk of various malignancies including breast cancer by meta-analyses [20]. However, this kind of meta-analyses was observational studies that showed limitations to evaluate plausible cause of this increased risk. We already showed peculiar trend of worse overall survival and breast cancer specific survival either in underweight or obese breast cancer patients [21]. Specifically, obese luminal A type patients showed worse survival, however, underweight HER2-amplifying patients exhibited worse survival outcome. Additionally, there seemed to be associated with menstrual status and BMI to induce breast cancer with different subtypes [22]. An increased BMI was one of the confounding factors to lead HER2-positive early breast cancer recurrence [23]. There was a review of modifiable risk factors inducing more breast cancers, which pointed out BMI might be one of the probable inducing factors, though lots of patients should be needed to obtain the statistical power [24]. Our data suggested that increased BMI may be associated with decreased IFN-γ concentration. Whether the IFN-γ concentration cut-off value was set at 100 or 250 pg/mL, there was a clear relationship between a decreased IFN-γ concentration and increased BMI. This correlation can be used as further evidence to support the association between increased BMI and decreased NK cell activity, which is already a poor prognostic factor. To date, no study has proposed that kind of possibility with observational findings from breast cancer patients.

There have been many reports published on systemic inflammatory markers including neutrophil, platelet, lymphocyte, monocyte, and eosinophil showing their association with worse clinical outcomes even in breast cancer [25,26,27]. All three ratios, namely N/L, P/N, and P/L, increased when the NK cell activity decreased to an abnormal level, compared to the other levels. To date, no other study evaluates these systemic inflammatory markers with peripheral NK cell activity. Our study is the first one to show positive relationships within peripheral NK cell activity with these three representative markers that are frequently studied for use as prognostic and predictive oncologic markers in various solid tumors. According to our study, along with other ones addressing the possible role of these serologic markers in prognosis prediction, one can more precisely estimate oncologic outcomes by adding NK activity to them. Other than BMI and inflammatory markers, we showed a statistically significant relationship between decreased IFN-γ concentration and advanced tumor staging. Despite its marginal significance, nodal involvement advanced as IFN-γ concentration decreased.

Differently from the findings of complex laboratory data analyzing subsets of NK cells, NK cell activity measurement by ELISA makes it pretty easy to obtain the value from the beginning at diagnosis of breast cancer [15]. Standard prognostic markers such as TNM and subtypes have been shown to have limitations in predicting the clinical course even after recommended standard systemic treatments; we believe NK cell activity measurement by ELISA can be used as adjunctive parameters to obtain more precise prognosis predictions. The other possible relationships between Ki-67 and molecular subtypes and/or nodal staging may be elucidated with a larger population and long-term follow up.

Some studies have already reported a correlation between NK cell activity, represented by interferon-gamma secretion, and benign diseases such as type II diabetes and herpes zoster, as well as smoking, unhealthy metabolic status, and physical inactivity [28,29,30]. There is also a study on the correlation between the neutrophil-to-lymphocyte ratio and NK cell activity in healthy populations, and the study showed that as the neutrophil-to-lymphocyte ratio increased, NK cell activity decreased [31].

In studies of hematologic malignancies, which are relatively more associated with immune function and more amenable to immunotherapy, it has been shown that an NK cell assay could be used as a screening test for hemophagocytic lymphohistiocytosis [32], and other studies have shown that NK cell activity is significantly lower in a variety of hematologic malignancies and can be used to monitor immunologic status, and that reduced NK cell activity is associated with the risk of developing cytomegalovirus disease after allogeneic-hematopoietic stem cell transplantation [33,34].

NK cell activity assays are also demonstrating their potential in high-risk screening, adjunctive role in diagnosis, prognosis prediction, and treatment outcome monitoring in clinical studies of patients with various solid tumors [15,35,36,37,38,39,40,41,42,43].

Chimeric antigen receptor (CAR) T-cell therapy has provided a new direction in the treatment of hematologic malignancies, but serious side effects including cytokine storm (cytokine release syndrome) have been reported, and unlike hematologic malignancies, its effectiveness in solid tumors appears to be limited, so efforts to identify biomarkers to monitor treatment effectiveness continue [44]. There are also reports that NK cell activity can be used to predict the effectiveness of immunotherapy in lung cancer, which is a relatively common disease [45,46].

In this context, to date, no CAR-based immunotherapeutic drug has been approved by the FDA for use in solid tumor treatments. More recently, CAR NK cell therapy, which is reported to have fewer side effects than CAR T-cell therapy, has been studied extensively, including early clinical studies in triple-negative breast cancer, which has shown the worst prognoses in breast cancer of all molecular subtypes and for which there are still no highly effective treatments available [47]. In conjunction with these efforts, an accurate, objective, effective, and simple test to select patients who are expected to respond optimally would help maximize the effectiveness while minimizing side effects of immunotherapy as well as other systemic treatments.

In recent years, attempts have been made to improve treatment outcomes by combining immunotherapeutic agents with conventional therapy in the triple-negative subtype of breast cancer, which is recognized as having the worst prognosis. However, breast cancer is well known as one of the less immunogenic cancers, so selecting the right patients is crucial to ensure that efforts are fruitful. However, many of the methods used to date have relied on tumor-infiltrating lymphocytes in tissue or complex, time-consuming, and expensive tests to assess patients’ immune status. Therefore, it is important to find a relatively accurate and inexpensive test that correlates well with established prognostic factors. In order to find such a method, this study was a pilot study to investigate the correlation with existing prognostic factors using an ELISA kit that can quantify peripheral NK cell activity by IFN-γ concentration at the time of diagnosis relatively easily and at low cost, with the limitations of being a retrospective study.

## 4. Materials and Methods

### 4.1. Study Population

This study was approved by the Institutional Review Board of our institution (approval number: VC20WISI0035). All procedures performed in this study involving human participants were in accordance with the ethical standards of the institutional and/or national research committee and conducted according to the guidelines of the Declaration of Helsinki in 1964 and its later amendments or comparable ethical standards. The electronic medical records were reviewed for breast cancer patients who received curative surgical treatment from March 2017 to July 2021 at the Department of Surgery at the Catholic University of Korea St. Vincent’s Hospital. In total, 1149 consecutive patients who received curative surgical treatment were initially recruited for this study. A total of 753 patients were excluded to minimize possible confounding factors, including 337 patients without NK cell activity result at diagnosis, 278 patients treated with neoadjuvant chemotherapy, 100 patients with ductal carcinoma, and 38 patients with bilateral breast cancer. More than 200 patients with preoperative systemic neoadjuvant chemotherapy were excluded from this study, mainly because peripheral NK cell activity was measured at diagnosis before commencing systemic chemotherapy that may affect peripheral NK cell activity [14]. A total of 396 patients were included and evaluated in this study (Table 5). This is a retrospective study, and 396 patients were divided into four groups based on NK cell activation-induced interferon-gamma (IFN-γ) secretion, with no difference in age or menopausal status among all groups. The four groups did not differ in terms of index tumor size and axillary lymph node involvement, which is clinically important for prognosis and disease severity. However, IFN-γ secretion tended to decrease with an increasing stage, and body mass index increased with a decreasing IFN-γ secretion, so we performed additional subgroup analyses on these factors to ensure that their effects did not bias the final results. We reviewed patients’ demographics and tumor characteristics including age, menopausal status, body surface area (BSA), body mass index (BMI) (kg/m^2^), type of surgery, pathological T and N staging, breast cancer stage according to the seventh edition of American Joint Committee on Cancer (AJCC) classification, histologic grade and type, lymphovascular invasion, Ki-67 proliferation index, estrogen receptor (ER), progesterone receptor (PR), and human epidermal growth factor receptor 2 (HER2) expression. ER and PR status was determined immunohistochemically and reported in the patients’ medical records. Immunohistochemistry (IHC), fluorescence in situ hybridization (FISH), or silver in situ hybridization (SISH) were used to evaluate HER2 status, and an IHC score of 0 or IHC score of 1+/2+ with negative FISH or SISH were defined as without HER2 overexpression. All patients were categorized into four subgroups according to the expression of ER, PR, and HER2 as follows: (a) ER and/or PR+/HER2− (luminal A group); (b) ER and/or PR+/HER2+ (luminal B group); (c) ER and PR−/HER2+ (HER2 group); (d) ER and PR−/HER2− (triple-negative group). The index tumor location was categorized according to the quadrant of the breast in which the cancer was located as follows: UIQ (upper inner quadrant), UOQ (upper outer quadrant), LIQ (lower inner quadrant), LOQ (lower outer quadrant), and SA (subareolar).

### 4.2. NK-Induced IFN-γ Secretion Assay to Determine NK Cell Activity

Peripheral natural killer cell activity was evaluated using a quantitative sandwich ELISA (enzyme-linked immunosorbent assay) kit to measure the released interferon-γ (IFN-γ) from natural killer cells to quantify NK cell activity. NK-induced IFN-γ secretion assay to determine NK cell activity was performed by ELISA using NK Vue-Kit (NKMAX^®^, Seongnam, Republic of Korea). Fresh whole blood (1 mL) was obtained using tubes containing Promoca (NKMAX^®^, Seongnam, Republic of Korea). Promoca is a stimulatory cytokine that can specifically stimulate NK cells. The main cell population secreting IFN-γ after stimulating whole blood with Promoca was NK cells. After incubation at 37 °C for 20~24 h, the samples were centrifuged at 11,500× *g* for 1 min, and the supernatant was transferred to a 1.5 mL microtube, which was then stored at −20 °C until of IFN-γ levels reached the recommended amount according to the manufacturer’s instructions. Briefly, 50 μL of six standards, controls, and samples was incubated in an antihuman IFN γ-coated plate at room temperature for 2 h and washed with washing buffer. IFN-γ conjugate was added and further incubated at room temperature for 1 h. After washing and incubation with 100 μL of the substrate at room temperature for 30 min in the dark, the absorbance value was measured at 450 nm. Concentrations of IFN-γ were determined with a calibration curve. The measuring range was 40~2000 pg/mL and the total imprecision for two levels of controls was less than the 15% coefficient of variations. Total range of measured concentration of IFN-γ was divided in quadripartite to be translated into 4 categories per manufacturer’s guideline. A measured value of 500 pg/mL or more is normal, a value of 250 and more but less than 500 pg/mL is concerned, a value of 100 and more but less than 250 is borderline, and a value less than 100 pg/mL is abnormal [15].

We also evaluated serum inflammatory markers such as neutrophil-to-eosinophil ratio (N/E), eosinophil-to-neutrophil ratio (E/N), monocyte-to-lymphocyte ration (M/L), lymphocyte-to-monocyte ratio (L/M), neutrophil-to-lymphocyte ratio (N/L), platelet-to-neutrophil ratio (P/N), and platelet-to-lymphocyte ratio (P/L), because there are several reports on their clinical correlation with major prognostic and predictive markers in breast cancer as well as other solid tumors.

### 4.3. Statistical Analysis

Categorical variables were reported as the number and percentages, and continuous variables were reported with mean ± standard deviation. The normality of distribution of continuous variables was tested by the Shapiro–Wilk or Kolmogorov–Smirnov test, and variance equality was assessed by Levene’s test. The comparison of continuous variables between groups was assessed using the student’s *t*-test, Mann–Whitney U-test, or one-way analysis of variance (ANOVA) with a Bonferroni post hoc comparison. The chi-square of Fisher’s exact test was used in categorical variables to assess the relationship between groups. Statistical analysis was performed using SPSS for Windows version 17.0 and a *p*-value < 0.05 was accepted as statistically significant.

## 5. Conclusions

Admittedly, this study retrospectively evaluated possible clinical correlation of peripheral NK cell activity measured by induced interferon-gamma as a prognostic or predictive marker by comparing major clinicopathologic parameters at diagnosis. Even though many studies to date have focused on tumor-infiltrating NK cells or molecular work-up to elucidate its role in breast cancer oncogenesis, most of the questions still need to be answered. We understand that this study is the first one to determine the correlation of peripheral NK cell activity represented by IFN-γ concentration with major clinicopathologic parameters at diagnosis. Although breast cancer is thought to be relatively unresponsive to immunotherapy, immunotherapy is expected to be effective in triple-negative breast cancer, and there are reports that 17β-estradiol is involved in the activity of NK cells in hormone-dependent breast cancer. As these treatments are increasingly likely to be used for breast cancer in the future, it is meaningful to study whether a peripheral blood NK cell activity test, which is simpler and easier for pathologists to perform than tumor-infiltrating NK cell analysis, can be used objectively to predict prognosis and select patients for immunotherapy, once it is possible to determine a meaningful relationship between pretreatment NK cell activity and prognosis before starting full-scale treatment. These results alone do not support the immediate use of peripheral NK cell activity test results in conjunction with established clinicopathologic factors to predict immunotherapy indications or prognosis in breast cancer. Re-establishing appropriate cut-off values in a larger number of diverse patients, or identifying precise cut-off values for each stage, would be a step forward. Research must continue on how to identify the optimal patients for costly immunotherapy in triple-negative breast cancer, which currently has the poorest prognosis and for which effective treatments are still in development, and in advanced breast cancer patients with recurrence and metastasis, for whom conventional treatments have reached their limits. For the ELISA kit used in this study, one might assume that functional assays, including cytotoxicity, would be required to assess NK cell activity. However, several papers have shown that measuring IFN-γ production can be used as a surrogate marker of NK cell activity [31,33,36,41,46]. There are also a number of studies that support the feasibility of using whole blood to measure IFN-γ to quantify NK cell activity without isolating NK cells [15,28,30,35,37,39,40,42]. And several studies have validated that most of the IFN-γ measured using the ELISA kit used in this study is produced by NK cells [37,48,49], and others have validated the mechanism by which this ELISA kit can be used to quantify NK cell activity in whole blood [49]. In this study, NK cell activity quantitated from the ELISA kit was first analyzed using the four result bands used in many solid tumors in previous studies recommended by the kit manufacturer, and then further statistical analyses were performed to minimize bias due to the relationship with patient-related factors and the limitations of retrospective studies. Therefore, the ongoing research after this study is to identify the cut-off value of IFN-γ concentration that is most highly correlated with existing prognostic factors, and to present the results of the analyses in each stage and subtype and in relation to BMI using the new cut-off value in order to help to select proper candidates for further immunotherapy.

## Figures and Tables

**Figure 1 ijms-25-04561-f001:**
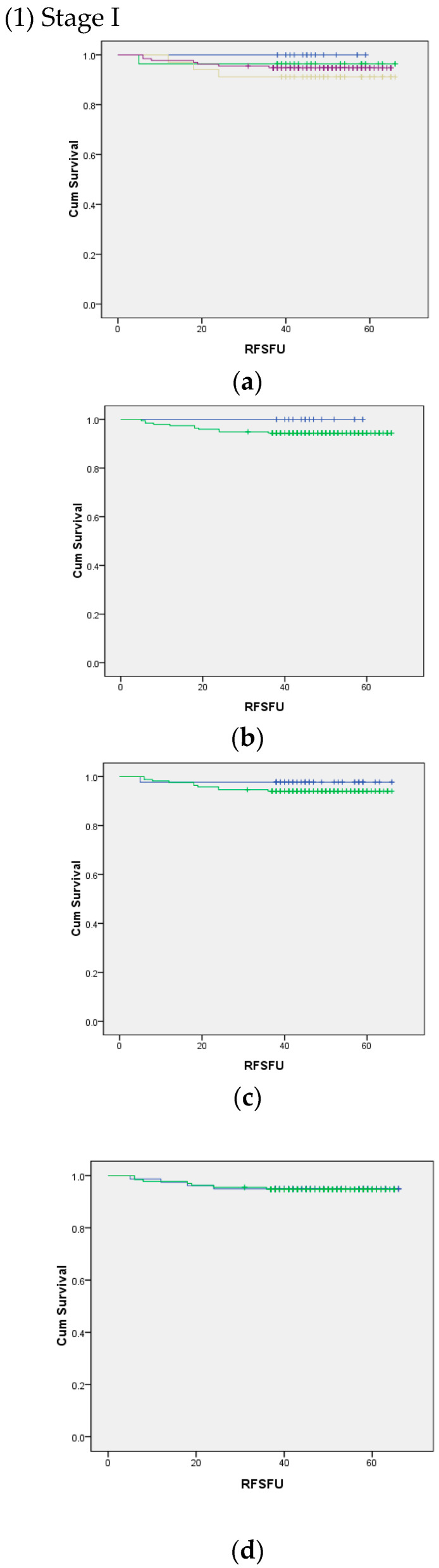
Log rank (Mantel–Cox) survival analyses of stage I to IV according to natural killer cell activity represented by interferon-gamma secretion (pg/mL).

**Table 1 ijms-25-04561-t001:** Binary subgroup analysis of natural killer (NK) cell activity by NK-induced interferon (IFN)-γ secretion (pg/mL).

(a) IFN-γ < 100 vs. IFN-γ ≥ 100				
		IFN-γ < 100	IFN-γ ≥ 100	*p*-value
Age	years	54.49 ± 10.496	53.18 ± 9.609	0.388
Menopause ^1^	No	21 (44.7%)	193 (55.3%)	0.170
	Yes	26 (55.3%)	156 (44.7%)	
Stage	0	0 (0%)	0 (0%)	<0.001
	1	17 (36.2%)	196 (56.2%)	
	2	17 (36.2%)	128 (36.7%)	
	3	12 (25.5%)	21 (6.0%)	
	4	1 (2.1%)	4 (1.1%)	
Tumor size	(cm)	2.0298 ± 1.63693	1.8232 ± 1.07634	0.404
T stage	1	32 (68.1%)	248 (71.1%)	0.131
	2	12 (25.5%)	95 (27.2%)	
	3	3 (6.4%)	6 (1.7%)	
Nodal involvement	−	28 (59.6%)	259 (74.2%)	0.035
	+	19 (40.4%)	90 (25.8%)	
N stage	0	28 (59.6%)	259 (74.2%)	0.007
	1	10 (21.3%)	66 (18.9%)	
	2	4 (8.5%)	18 (5.2%)	
	3	5 (10.6%)	6 (1.7%)	
BSA ^2^	(m^2^)	1.6172 ± 0.14379	1.6282 ± 0.25912	0.776
BMI ^3^	<18.5	2 (4.3%)	3 (0.9%)	0.033
	18.5~25	19 (40.4%)	204 (58.5%)	
	25~30	21 (44.7%)	110 (31.5%)	
	≥30	5 (10.6%)	32 (9.2%)	
Tumor location ^4^	UIQ	13 (27.7%)	93 (26.6%)	0.658
	UOQ	20 (42.6%)	150 (43.0%)	
	LIQ	4 (8.5%)	17 (4.9%)	
	LOQ	4 (8.5%)	52 (14.9%)	
	SA	6 (12.8%)	37 (10.6%)	
Subtype ^5^	Luminal A	30 (63.8%)	224 (64.2%)	0.129
	Luminal B	6 (12.8%)	55 (15.8%)	
	HER2	2 (4.3%)	38 (10.9%)	
	TNBC	9 (19.1%)	32 (9.2%)	
Ki-67	(%)	36.3987 ± 30.67887	31.2337 ± 27.44844	0.238
Inflammatory markers ^6^	N/L	2.616840 ± 1.8321762	1.920343 ± 1.5769598	0.016
	P/N	145.029801 ± 306.9068589	91.740120 ± 55.6915036	0.241
	P/L	181.947542 ± 83.1788952	146.214319 ± 58.1006509	0.006
	L/M	11.011792 ± 21.5371768	6.449008 ± 9.8979492	0.155
	M/L	0.257049 ± 0.1943889	0.231698 ± 0.2081499	0.430
	E/N	0.037087 ± 0.0496740	0.038719 ± 0.0363930	0.828
	N/E	81.737374 ± 76.9226519	65.807859 ± 100.6729953	0.297
(b) IFN-γ < 250 vs. IFN-γ ≥ 250.				
		IFN-γ < 250	IFN-γ ≥ 250	*p*-value
Age	years	52.7396 ± 10.19829	53.5300 ± 9.56316	0.488
Menopause ^1^	No	55 (57.3%)	159 (53.0%)	0.463
	Yes	41 (42.7%)	141 (47.0%)	
Stage	0	0 (0%)	0 (0%)	0.007
	1	45 (46.9%)	168 (56.0%)	
	2	33 (34.4%)	112 (37.3%)	
	3	16 (16.7%)	17 (5.7%)	
	4	2 (2.1%)	3 (1.0%)	
Tumor size	(cm)	1.9167 ± 1.38250	1.8257 ± 1.07575	0.556
T stage	1	67 (69.8%)	213 (71.0%)	0.811
	2	26 (27.1%)	81 (27.0%)	
	3	3 (3.1%)	6 (2.0%)	
Nodal involvement	−	62 (64.6%)	225 (75.0%)	0.047
	+	34 (35.4%)	75 (25.0%)	
N stage	0	62 (64.6%)	225 (75.0%)	0.111
	1	21 (21.9%)	55 (18.3%)	
	2	8 (8.3%)	14 (4.7%)	
	3	5 (5.2%)	6 (2.0%)	
BSA ^2^	(m^2^)	1.6229 ± 0.14403	1.6282 ± 0.27340	0.856
BMI ^3^	<18.5	3 (3.1%)	2 (0.7%)	0.038
	18.5~25	47 (49.0%)	176 (58.7%)	
	25~30	32 (33.3%)	99 (33.0%)	
	≥30	14 (14.6%)	23 (7.7%)	
Tumor location ^4^	UIQ	21 (21.9%)	85 (28.3%)	0.657
	UOQ	42 (43.8%)	128 (42.7%)	
	LIQ	7 (7.3%)	14 (4.7%)	
	LOQ	14 (14.6%)	42 (14.0%)	
	SA	12 (12.5%)	31 (10.3%)	
Subtype ^5^	Luminal A	61 (63.5%)	193 (64.3%)	0.809
	Luminal B	15 (15.6%)	46 (15.3%)	
	HER2	8 (8.3%)	32 (10.7%)	
	TNBC	12 (12.5%)	29 (9.7%)	
Ki-67	(%)	34.0012 ± 30.09100	31.1493 ± 27.12083	0.385
Inflammatory markers ^6^	N/L	2.405103 ± 1.4916710	1.874338 ± 1.6439278	0.005
	P/N	112.801350 ± 217.2852797	93.349244 ± 58.3223077	0.388
	P/L	170.527035 ± 70.9506157	144.032454 ± 58.3097807	0.001
	L/M	7.756715 ± 15.2508663	6.745379 ± 10.6270480	0.547
	M/L	0.251421 ± 0.1512812	0.229359 ± 0.2212231	0.363
	E/N	0.035809 ± 0.0438872	0.039393 ± 0.0361442	0.424
	N/E	88.588240 ± 106.8376801	60.981585 ± 94.4543589	0.025
(c) IFN-γ < 500 vs. IFN-γ ≥ 500.				
		IFN-γ < 500	IFN-γ ≥ 500	*p*-value
Age	years	52.4342 ± 9.91465	53.9016 ± 9.56366	0.144
Menopause ^1^	No	87 (57.2%)	127 (52.0%)	0.314
	Yes	65 (42.8%)	117 (48.0%)	
Stage	0	0 (0%)	0 (0%)	0.02
	1	79 (52.0%)	134 (54.9%)	
	2	50 (32.9%)	95 (38.9%)	
	3	19 (12.5%)	14 (5.7%)	
	4	4 (2.6%)	1 (0.4%)	
Tumor size	(cm)	1.8961 ± 1.29364	1.8176 ± 1.06364	0.512
T stage	1	110 (72.4%)	170 (69.7%)	0.392
	2	37 (24.3%)	70 (28.7%)	
	3	5 (3.3%)	4 (1.6%)	
Nodal involvement	−	101 (66.4%)	186 (76.2%)	0.034
	+	51 (33.6%)	58 (23.8%)	
N stage	0	101 (66.4%)	186 (76.2%)	0.202
	1	35 (23.0%)	41 (16.8%)	
	2	11 (7.2%)	11 (4.5%)	
	3	5 (3.3%)	6 (2.5%)	
BSA ^2^	(m^2^)	1.6188 ± 0.14011	1.6320 ± 0.29635	0.606
BMI ^3^	<18.5	4 (2.6%)	1 (0.4%)	0.254
	18.5~25	82 (53.9%)	141 (57.8%)	
	25~30	50 (32.9%)	81 (33.2%)	
	≥30	16 (10.5%)	21 (8.6%)	
Tumor location ^4^	UIQ	35 (23.0%)	71 (29.1%)	0.156
	UOQ	63 (41.4%)	107 (43.9%)	
	LIQ	10 (6.6%)	11 (4.5%)	
	LOQ	29 (19.1%)	27 (11.1%)	
	SA	15 (9.9%)	28 (11.5%)	
Subtype ^5^	Luminal A	91 (59.9%)	163 (66.8%)	0.563
	Luminal B	27 (17.8%)	34 (13.9%)	
	HER2	17 (11.2%)	23 (9.4%)	
	TNBC	17 (11.2%)	24 (9.8%)	
Ki-67	(%)	34.4825 ± 28.95429	30.1969 ± 27.07719	0.137
Inflammatory markers ^6^	N/L	2.255324 ± 1.2675602	1.845828 ± 1.7934494	0.014
	P/N	100.654264 ± 174.6781224	96.451863 ± 61.5558317	0.732
	P/L	161.793037 ± 65.5709637	143.392582 ± 59.6611772	0.004
	L/M	7.524051 ± 13.5422745	6.658207 ± 10.7678567	0.482
	M/L	0.238380 ± 0.1406386	0.232419 ± 0.2387599	0.780
	E/N	0.034561 ± 0.0396289	0.040993 ± 0.0370471	0.103
	N/E	81.968786 ± 105.9055164	58.846488 ± 92.2006054	0.028

^1^ Menopause was defined as either blood FSH (follicle-stimulating hormone) over 30 mIU/mL and not having had a menstrual period for a year or bilateral oophorectomy for any reason. ^2^ BSA (body surface area) (m^2^) by DuBois formula. ^3^ BMI (body mass index) (kg/m^2^). ^4^ The index tumor location was categorized according to the quadrant of the breast in which the cancer was located as follows: UIQ (upper inner quadrant), UOQ (upper outer quadrant), LIQ (lower inner quadrant), LOQ (lower outer quadrant), and SA (subareolar). ^5^ Molecular subtypes were categorized into four subgroups according to the expression of ER, PR, and HER2 as follows: (a) ER and/or PR+/HER2− (luminal A group); (b) ER and/or PR+/HER2+ (luminal B group); (c) ER and PR−/HER2+ (HER2 group); (d) ER and PR−/HER2− (triple-negative group). ^6^ Serum inflammatory markers; neutrophil-to-eosinophil ratio (N/E), eosinophil-to-neutrophil ratio (E/N), monocyte-to-lymphocyte ration (M/L), lymphocyte-to-monocyte ratio (L/M), neutrophil-to-lymphocyte ratio (N/L), platelet-to-neutrophil ratio (P/N), and platelet-to-lymphocyte ratio (P/L).

**Table 2 ijms-25-04561-t002:** ANOVA post hoc test (Bonferroni).

Inflammatory Markers	Concentration of IFN-γ (pg/mL)				*p*-Value	*p*-Value
	IFN-γ < 100	100 ≤ IFN-γ < 250	250 ≤ IFN-γ < 500	IFN-γ ≥ 500		
N/L	2.617 ± 1.83	2.202 ± 1.05	1.999 ± 0.68	1.846 ± 1.79	0.02	
	①	②				1.000
	①		③			0.316
	①			④		0.017
		②	③			1.000
		②		④		0.948
			③	④		1.000
P/N	145.030 ± 306.91	81.888 ± 34.39	79.831 ± 39.01	96.452 ± 61.56	0.021	
	①	②				0.052
	①		③			0.031
	①			④		0.058
		②	③			1.000
		②		④		1.000
			③	④		1.000
P/L	181.948 ± 83.18	159.573 ± 55.52	146.820 ± 52.42	143.393 ± 59.66	0.001	
	①	②				0.453
	①		③			0.025
	①			④		0.001
		②	③			1.000
		②		④		0.562
			③	④		1.000
L/M	11.012 ± 21.54	4.634 ± 1.70	7.125 ± 10.07	6.658 ± 10.77	0.057	
	①	②				0.052
	①		③			0.586
	①			④		0.128
		②	③			1.000
		②		④		1.000
			③	④		1.000
M/L	0.257 ± 0.19	0.246 ± 0.95	0.216 ± 0.12	0.232 ± 0.24	0.758	
	①	②				1.000
	①		③			1.000
	①			④		1.000
		②	③			1.000
		②		④		1.000
			③	④		1.000
E/N	0.037 ± 0.05	0.035 ± 0.04	0.032 ± 0.03	0.041 ± 0.04	0.386	
	①	②				1.000
	①		③			1.000
	①			④		1.000
		②	③			1.000
		②		④		1.000
			③	④		0.780
N/E	81.737 ± 76.92	95.159 ± 129.73	70.415 ± 104.22	58.846 ± 92.20	0.123	
	①	②				1.000
	①		③			1.000
	①			④		0.854
		②	③			1.000
		②		④		0.109
			③	④		1.000

**Table 3 ijms-25-04561-t003:** Subgroup analysis by TNM Stage with Bonferroni adjustment.

Stage	Concentration of IFN-γ (pg/mL)				*p*-Value	*p*-Value
	IFN-γ < 100	100 ≤ IFN-γ < 250	250 ≤ IFN-γ < 500	IFN-γ ≥ 500		
I (n, %)	17/47 (36.2%)	28/49 (57.1%)	34/56 (60.7%)	134/244 (54.9%)	0.063	
	①	②				0.04
	①		③			0.013
	①			④		0.018
		②	③			0.71
		②		④		0.775
			③	④		0.431
II (n, %)	17/47 (36.2%)	16/49 (32.7%)	17/56 (30.4%)	95/244 (38.9%)	0.605	
	①	②				0.717
	①		③			0.532
	①			④		0.721
		②	③			0.8
		②		④		0.408
			③	④		0.231
III (n, %)	12/47 (25.5%)	4/49 (8.2%)	3/56 (5.4%)	13/244 (5.3%)	<0.001	
	①	②				0.022
	①		③			0.004
	①			④		<0.001
		②	③			0.425
		②		④		0.311
			③	④		0.602
IV (n, %)	1/46 (2.1%)	1/48 (2%)	2/56 (3.6%)	1/244 (0.4%)	0.097	
	①	②				0.742
	①		③			0.566
	①			④		0.297
		②	③			0.55
		②		④		0.307
			③	④		0.091

**Table 4 ijms-25-04561-t004:** Subgroup analysis by BMI with Bonferroni adjustment.

(a) BMI < 18.5 vs. 18.5 ≤ BMI < 25 vs. 25 ≤ BMI < 30 vs. BMI ≥ 30						
Body mass index (kg/m^2^)	Concentration of IFN-γ (pg/mL)				*p*-value	*p*-value
	IFN-γ < 100	100 ≤ IFN-γ < 250	250 ≤ IFN-γ < 500	IFN-γ ≥ 500		
<18.5	2/47 (4.3%)	1/49 (2%)	1/56 (1.8%)	1/244 (0.4%)	0.066	
	①	②				0.484
	①		③			0.434
	①			④		0.069
		②	③			0.718
		②		④		0.307
			③	④		0.339
18.5 ≤ BMI < 25	19/47 (40.4%)	28/49 (57.1%)	35/56 (62.5%)	141/244 (57.8%)	0.115	
	①	②				0.101
	①		③			0.025
	①			④		0.028
		②	③			0.576
		②		④		0.934
			③	④		0.518
25 ≤ BMI < 30	21/47 (44.7%)	11/49 (22.4%)	18/56 (32.1%)	81/244 (33.2%)	0.146	
	①	②				0.021
	①		③			0.191
	①			④		0.131
		②	③			0.268
		②		④		0.139
			③	④		0.88
≥30	5/47 (10.6%)	9/49 (18.4%)	2/56 (3.6%)	21/244 (8.6%)	0.075	
	①	②				0.283
	①		③			0.153
	①			④		0.413
		②	③			0.014
		②		④		0.04
			③	④		0.159
(b) BMI < 18.5 vs. 18.5 ≤ BMI < 25 vs. BMI ≥ 25						
Body mass index (kg/m^2^)	Concentration of IFN-γ (pg/mL)				*p*-value	*p*-value
	IFN-γ <100	100≤ IFN-γ < 250	250≤ IFN-γ < 500	IFN-γ ≥ 500		
<18.5	2/47 (4.3%)	1/49 (2%)	1/56 (1.8%)	1/244 (0.4%)	0.066	
	①	②				0.484
	①		③			0.434
	①			④		0.069
		②	③			0.718
		②		④		0.307
			③	④		0.339
18.5 ≤ BMI < 25	19/47 (40.4%)	28/49 (57.1%)	35/56 (62.5%)	141/244 (57.8%)	0.115	
	①	②				0.101
	①		③			0.025
	①			④		0.028
		②	③			0.576
		②		④		0.934
			③	④		0.518
BMI ≥ 25	26/47 (55.3%)	20/49 (40.8%)	20/56 (35.7%)	102/244 (41.8%)	0.229	
	①	②				0.155
	①		③			0.046
	①			④		0.087
		②	③			0.591
		②		④		0.898
			③	④		0.403
(c) BMI < 25 vs. 25 ≤ BMI < 30 vs. BMI ≥ 30						
Body mass index (kg/m^2^)	Concentration of IFN-γ (pg/mL)				*p*-value	*p*-value
	IFN-γ < 100	100 ≤ IFN-γ < 250	250≤ IFN-γ < 500	IFN-γ ≥ 500		
<25	21/47 (44.7%)	29/49 (59.2%)	36/56 (64.3%)	142/244 (58.2%)	0.229	
	①	②				0.155
	①		③			0.046
	①			④		0.087
		②	③			0.591
		②		④		0.898
			③	④		0.403
25 ≤ BMI < 30	21/47 (44.7%)	11/49 (22.4%)	18/56 (32.1%)	81/244 (33.2%)	0.146	
	①	②				0.021
	①		③			0.191
	①			④		0.131
		②	③			0.268
		②		④		0.139
			③	④		0.88
BMI ≥ 30	5/47 (10.6%)	9/49 (18.4%)	2/56 (3.6%)	21/244 (8.6%)	0.075	
	①	②				0.283
	①		③			0.153
	①			④		0.413
		②	③			0.014
		②		④		0.04
			③	④		0.159

**Table 5 ijms-25-04561-t005:** Patients’ demographics stratified by natural killer (NK) cell activity by NK-induced interferon (IFN)-γ secretion (pg/mL).

	(pg/mL)	IFN-γ < 100	100 ≤ IFN-γ < 250	250 ≤ IFN-γ < 500	IFN-γ ≥ 500	*p*-Value
Age	years	54.49 ± 10.496	51.06 ± 9.716	51.91 ± 9.476	53.90 ± 9.564	0.146
Menopause ^1^	No	21 (44.7%)	34 (69.4%)	32 (57.1%)	127 (52.0%)	0.075
	Yes	26 (55.3%)	15 (30.6%)	24 (42.9%)	117 (48.0%)	
Stage	0	0 (0%)	0 (0%)	0 (0%)	0 (0%)	0.001
	1	17 (36.2%)	28 (57.1%)	34 (60.7%)	134 (54.9%)	
	2	17 (36.2%)	16 (32.7%)	17 (30.4%)	95 (38.9%)	
	3	12 (25.5%)	4 (8.2%)	3 (5.4%)	14 (5.7%)	
	4	1 (2.1%)	1 (2.0%)	2 (3.6%)	1 (0.4%)	
Tumor size	(cm)	2.0298 ± 1.63693	1.8082 ± 1.09103	1.8607 ± 1.13629	1.8176 ± 1.06364	0.708
T stage	0	0 (0%)	0 (0%)	0 (0%)	0 (0%)	0.306
	1	32 (68.1%)	35 (71.4%)	43 (76.8%)	170 (69.7%)	
	2	12 (25.5%)	14 (28.6%)	11 (10.3%)	70 (28.7%)	
	3	3 (6.4%)	0 (0%)	2 (3.6%)	4 (1.6%)	
Axillary node involvement	−	21 (44.7%)	34 (69.4%)	32 (57.1%)	127 (52.0%)	0.075
	+	26 (55.3%)	15 (30.6%)	24 (42.9%)	117 (48.0%)	
N stage	0	28 (59.6%)	34 (69.4%)	39 (69.6%)	186 (76.2%)	0.057
	1	10 (21.3%)	11 (22.4%)	14 (25.0%)	41 (16.8%)	
	2	4 (8.5%)	4 (8.2%)	3 (5.4%)	11 (4.5%)	
	3	5 (10.6%)	0 (0%)	0 (0%)	6 (2.5%)	
BSA ^2^	(m^2^)	1.6172 ± 0.14379	1.6284 ± 0.14554	1.6116 ± 0.13411	1.6320 ± 0.29635	0.943
BMI ^3^	<18.5	2 (4.3%)	1 (2.0%)	1 (1.8%)	1 (0.4%)	0.021
	18.5~25	19 (40.4%)	28 (57.1%)	35 (62.5%)	141 (57.8%)	
	25~30	21 (44.7%)	11 (22.4%)	18 (32.1%)	81 (33.2%)	
	≥30	5 (10.6%)	9 (18.4%)	2 (3.6%)	21 (8.6%)	
Tumor location ^4^	UIQ	13 (27.7%)	8 (16.3%)	14 (25.0%)	71 (29.1%)	0.166
	UOQ	20 (42.6%)	22 (44.9%)	21 (37.5%)	107 (43.9%)	
	LIQ	4 (8.5%)	3 (6.1%)	3 (5.4%)	11 (4.5%)	
	LOQ	4 (8.5%)	10 (20.4%)	15 (26.8%)	27 (11.1%)	
	SA	6 (12.8%)	6 (12.2%)	3 (5.4%)	28 (11.5%)	
Molecular subtype ^5^	Luminal A	30 (63.8%)	31 (63.3%)	30 (53.6%)	163 (66.8%)	0.223
	Luminal B	6 (12.8%)	9 (18.4%)	12 (21.4%)	34 (13.9%)	
	HER2	2 (4.3%)	6 (12.2%)	9 (16.1%)	23 (9.4%)	
	TNBC	9 (19.1%)	3 (6.1%)	5 (8.9%)	24 (9.8%)	
Ki-67 index	%	36.3987 ± 30.67887	31.7504 ± 29.66749	35.2989 ± 27.16258	30.1969 ± 27.07719	0.401
Inflammatory markers ^6^	N/L	2.616840 ± 1.8321762	2.202012 ± 1.0491911	1.998554 ± 0.6793654	1.845830 ± 1.7934508	0.02
	P/N	145.029801 ± 306.9068589	81.888347 ± 34.3870716	79.830690 ± 39.0120772	96.451865 ± 61.5558313	0.021
	P/L	181.947542 ± 83.1788952	159.572676 ± 55.5197964	146.820472 ± 52.4189630	143.392587 ± 59.6611767	0.001
	L/M	11.011792 ± 21.5371768	4.634485 ± 1.6977871	7.125201 ± 10.0745587	6.658205 ± 10.7678564	0.057
	M/L	0.257049 ± 0.1943889	0.246025 ± 0.949740	0.216026 ± 0.1181874	0.232418 ± 0.2387606	0.758
	E/N	0.037087 ± 0.0496740	0.034598 ± 0.0379960	0.032422 ± 0.0312608	0.040996 ± 0.0370472	0.386
	N/E	81.737374 ± 76.9226519	95.159482 ± 129.7312489	70.414836 ± 104.2158182	58.846490 ± 92.2006053	0.123

^1^ Menopause was defined as either blood FSH (follicle-stimulating hormone) over 30 mIU/mL and not having had a menstrual period for a year or bilateral oophorectomy for any reason. ^2^ BSA (body surface area) (m^2^) by DuBois formula. ^3^ BMI (body mass index) (kg/m^2^). ^4^ The index tumor location was categorized according to the quadrant of the breast in which the cancer was located as follows: UIQ (upper inner quadrant), UOQ (upper outer quadrant), LIQ (lower inner quadrant), LOQ (lower outer quadrant), and SA (subareolar). ^5^ Molecular subtypes were categorized into four subgroups according to the expression of ER, PR and HER2 as follows: (a) ER and/or PR+/HER2− (luminal A group); (b) ER and/or PR+/HER2+ (luminal B group); (c) ER and PR−/HER2+ (HER2 group), and (d) ER and PR−/HER2− (triple-negative group). ^6^ Serum inflammatory markers; neutrophil-to-eosinophil ratio (N/E), eosinophil-to-neutrophil ratio (E/N), monocyte-to-lymphocyte ration (M/L), lymphocyte-to-monocyte ratio (L/M), neutrophil-to-lymphocyte ratio (N/L), platelet-to-neutrophil ratio (P/N), and platelet-to-lymphocyte ratio (P/L).

## Data Availability

The data presented in this study may be available on request from the corresponding author. The data are not publicly available due to strict privacy protection act of the Republic of Korea.

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
