# Peer review of "Interferon-Gamma Secretion Is Significantly Decreased in Stage III Breast Cancer Patients"

_ijms, 2024, doi:10.3390/ijms25084561_

Round 1
Reviewer 1 Report (New Reviewer)
Comments and Suggestions for Authors
Suggestion:
Abstract:
1. Authors are suggested to focus on significant results of their study. P-values should be added along with results.
Introduction:
2. Authors should clearly state what the authors aim to investigate and improve it by providing a more concise background. It is suggested for authors to mention specific immunotherapeutic agents and their outcomes in breast cancer trials.
Materials and Methods:
3. The ELISA method used for measuring NK cell activity is briefly mentioned. Authors are suggested to provide more details about the specific reagents used, incubation conditions and standard curve preparation. Moreover, patient related factors (n=396) such as age distribution, menopausal status distribution and the distribution of different breast cancer stages should be mentioned.
Discussion:
4. Authors should strengthen the discussion by providing a more thorough interpretation of the results by inter-connecting the findings to the existing literature. The discussion mentioned the discrepancy between findings and previous research on NK cell activity. Authors should clarify the possible logical reasons for these differences and must discuss potential factors contributing to variations in results.
5. Authors are suggested to add specific recommendations for future research and also provide limitations of the current study
Conclusion:
6. Authors mentioned the need for re-establishing cut-off values and continuing research. Authors should be more specific about the steps that should be taken in future research to enhance the clinical applicability of peripheral NK cell activity testing.
7. Abbreviations list should be added.
Author Response
Suggestion:
Abstract:
- Authors are suggested to focus on significant results of their study. P-values should be added along with results.
I've rewritten the result part of the abstract as you suggested (line 9, lines 18-39, lines 42-50).
Introduction:
- Authors should clearly state what the authors aim to investigate and improve it by providing a more concise background. It is suggested for authors to mention specific immunotherapeutic agents and their outcomes in breast cancer trials.
As you recommended, we have inserted the names of immunotherapeutic agents currently being tried in lines 59-60, clarified the conflicting literature on the role of tumor infiltrating lymphocytes (TILs) in lines 92-93, and revised the text in lines 94-107 to provide a rationale for the purpose of this study and what we expect to learn.
Materials and Methods:
- The ELISA method used for measuring NK cell activity is briefly mentioned. Authors are suggested to provide more details about the specific reagents used, incubation conditions and standard curve preparation.
The ELISA kit used in this study was the NK Vue-Kit, which is patented by NKMAX (NKMAX®, Seongnam, Korea), and all procedures were performed according to the manufacturer's protocol, and all procedures were recorded verbatim from the manufacturer.
Moreover, patient related factors (n=396) such as age distribution, menopausal status distribution and the distribution of different breast cancer stages should be mentioned.
Because this is a retrospective study rather than a prospective randomized study, we cannot completely exclude the possibility that patient-related factors may have influenced the NK cell activity in each group, but we did subgroup analyses to minimize and prevent bias from patient related factors. All patient related factors are recorded in Table 1. and the p values for each group of NK cell activity are also recorded. (line 125~132)
Discussion
- Authors shouldstrengthen the discussion by providing a more thorough interpretation of the results by inter-connecting the findings to the existing literature. The discussion mentioned the discrepancy between findings and previous research on NK cell activity. Authors should clarify the possible logical reasons for these differences and must discuss potential factors contributing to variations in results.
As you pointed out, we have expanded the Discussion to include more information about the findings of this study and our reasoning for the differences from previous studies (lines 390-431, 448-452).
- Authors are suggested toadd specific recommendations for future research and also provide limitations of the current study
I added up recommendation and limitation at the end of discussion per recommendation. (line 508-519).
Conclusion:
- Authors mentioned the need for re-establishing cut-off values and continuing research. Authors should be more specific about the steps that should be taken in future research to enhance the clinical applicability of peripheral NK cell activity testing.
Per recommendation, I added up these remarks at the end of conclusion, per recommendation. (line 544-552)
- Abbreviations list should be added.
I attached that list at line 680-703.
Reviewer 2 Report (New Reviewer)
Comments and Suggestions for Authors
The Authors undetook an interesting topic to correlate IFNgamma production by NK cells in breast cancer patients, trying to correlate these results with the stage of a cancer.
It is very surprising to me that there are a lot of uncorrected changes in the manuscript, including the title, although this is the very first round of review. The title is still not corresponding to what was really analyzed, as NK cell activity corresponds to functional assays, including cytotoxicity, not only one cytokine production. Moreover, stimulation of whole blood cells does not mean IFN was produced by only NK cell. Here, this method part also sound suspicious, as I cannot image whole blood sample is kept for 24 hours, in a cell incubator? Why PBMC or NK cells were not isolated? The methods are difficult to be understood, especially this part is not divided by subheadings.
In sum, the paper present very low scientific soundness and cannot be accepted.
Comments on the Quality of English LanguageMust be imrpoved
Author Response
The Authors undetook an interesting topic to correlate IFNgamma production by NK cells in breast cancer patients, trying to correlate these results with the stage of a cancer.
It is very surprising to me that there are a lot of uncorrected changes in the manuscript, including the title, although this is the very first round of review.
First of all, I apologize for wasting your valuable time with a manuscript that appears to be disorganized. However, I would like to respectfully apologize for the fact that this manuscript was sent to you by manuscript editor in the middle of revision, when three other reviewers responded with revisions before your review round and the manuscript editor requested a resubmission.
The title is still not corresponding to what was really analyzed, as NK cell activity corresponds to functional assays, including cytotoxicity, not only one cytokine production.
The titles are rewritten according to the results obtained. The content is a study that evaluated the NK cell activity correlated with the amount of interferon-gamma produced to determine the correlation with existing breast cancer prognostic factors.
Moreover, stimulation of whole blood cells does not mean IFN was produced by only NK cell. Here, this method part also sound suspicious, as I cannot image whole blood sample is kept for 24 hours, in a cell incubator? Why PBMC or NK cells were not isolated? The methods are difficult to be understood, especially this part is not divided by subheadings.
Per recommendation, I separated that part under title of NK-induced IFN-γ secretion assay to determine NK cell activity (line 150) The method used in this study was not developed by me, but the EISA kit and all reagents used were manufactured by NKMAX (NKMAX®, Seongnam, Korea) based on her patents. Therefore, the process described in the manuscript was also disclosed by the manufacturer, and it was confirmed in PUBMED that the same process has already been published in several papers on various solid tumors and benign diseases. (ref. #15)

Round 2
Reviewer 2 Report (New Reviewer)
Comments and Suggestions for Authors
Although corrections, the title does still not fit the content - the paper merely descibe IFN production by PBMC, not NK cell activity. NK cell activity has not been evaluated as no NK cell functional assays were perfomed. Until the paper is thoroughly corrected taking this into concideration, I cannot support publication of this manuscript.
Comments on the Quality of English LanguageMinor mistakes were detected.
Author Response
Reviewer #2.
Comments and Suggestions for Authors
The Authors undetook an interesting topic to correlate IFNgamma production by NK cells in breast cancer patients, trying to correlate these results with the stage of a cancer.
The title is still not corresponding to what was really analyzed, as NK cell activity corresponds to functional assays, including cytotoxicity, not only one cytokine production.
The titles are rewritten according to the results obtained. As your recommendation, I replaced NK cell activity to IFN-gamma secretion, The content is a study that evaluated the NK cell activity correlated with the amount of interferon-gamma produced to determine the correlation with existing breast cancer prognostic factors.
I had attached five references those support that measurement of IFNgamma production (NK Vue ELISA kit) can be used as surrogate marker of NK cell activity.
- Kim BR et al. Association of neutrophil-to-lymphocyte ratio and natural killer cell activity revealed by measurement of interferon-gamma levels in a healthy population. J Clin Lab Anal2019;33(1):e22640. doi: 10.1002/jcla.22640.
- Park SH et al. Variable natural killer cell activity in hematologicalmalignancies at diagnosis. Lab Med Online2018;8(2):41-51.
- Choi SI et al. Clinical utility of a novel natural killer cell activity assay for diagnosing non-small cell lung cancer: a prospective pilot study. Onco Targets Ther2019;12:1661-9.
- Hansen TF et al. Correlation between natural killer cell activity and treatment in pateints with disseminated cancer. Transl Oncol2019;12(7):968-72.
- Choi MG et al. Efficacy of natural killer cell activity as a biomarker for predicting immunotherapy response in non-small cell lung cancer. Thorac Cancer2020;11(11):3337-45.
Moreover, stimulation of whole blood cells does not mean IFN was produced by only NK cell. Here, this method part also sound suspicious, as I cannot image whole blood sample is kept for 24 hours, in a cell incubator? Why PBMC or NK cells were not isolated? The methods are difficult to be understood, especially this part is not divided by subheadings.
Per recommendation, I separated that part under title of NK-induced IFN-γ secretion assay to determine NK cell activity (line 150) The method used in this study was not developed by me, but the EISA kit and all reagents used were manufactured by NKMAX (NKMAX®, Seongnam, Korea) based on her patents. Therefore, the process described in the manuscript was also disclosed by the manufacturer, and it was confirmed in PUBMED that the same process has already been published in several papers on various solid tumors and benign diseases. (ref. #15)
Although corrections, the title does still not fit the content - the paper merely descibe IFN production by PBMC, not NK cell activity. NK cell activity has not been evaluated as no NK cell functional assays were perfomed. Until the paper is thoroughly corrected taking this into concideration, I cannot support publication of this manuscript.
I had attached 8 references those support the method of measurement of IFNgamma from whole blood to measure NK cell activity.
- Kim JH et al. Relationship between natural killer cell activity and glucose control in patients with type 2 diabetes and prediabetes. J Diabetes Investig2019;10(5):1223-8.
- Jung YS et al. Physical inactivity and unhealthy metabolic status are associated with decreased natural killer cell activity. Yonsei Med J2018;59(4):554-62.
- Jobin G et al. Association between natural killer cell activity and colorectal cancer in high-risk subjects undergoing colonoscopy. Gastroenterology2017;153(4):980-7.
- Barkin et al. Association between natural killer cell activity and prostate cancer: a pilot study. Can J Urol2017;24(2):8708-13.
- Lee SB et al. A high-throughput assay of NK cell activity in whole blood and its clinical application. Biochem Biophys Res Commun2014;445(3):584-90.
- Lee J et al. Natural killer cell activity for IFN-gamma production as a supportive diagnostic marker for gastric cancer. Oncotarget2017;8(41):70431-40.
- Lee HS. Peripheral natural killer cell activity is associated with poor clinical outcomes in pancreatic ductal adenocarcinoma. J Gastroenterol Hepatol2021;36(2):516-22.
- Angka L. Natural killer cell IFNγsecretion is profoundly suppressed following colorectal cancer surgery. Ann Surg Oncol 2018;25(12):3747-54.
And, there are 1 reference and 2 additional articles that supports most of IFNgamma measured by NK Vue ELISA kit is secreted by NK cells.
- Lee SB et al. A high-throughput assay of NK cell activity in whole blood and its clinical application. Biochem Biophys Res Commun2014;445(3):584-90.
Koo KC et al. Reduction of the CD16(-)CD56bright NK cell subset precedes NK cell dysfunction in prostate cancer. PLoS One. 2013 Nov 4;8(11):e78049.
Nederby L. et al. Quantification of NK cell activity using whole blood: Methodological aspects of a new test. J Immunol Methods. 2018 Jul;458:21-25.
Lastly, the last article by Nederby L. et al. had proven the mode of action of NK Vue ELISA kit for quantification of NK cell activity.
I added up the background of the use of the ELISA kit in this study to use quantified IFN-gamma secretion by NK cell from whole blood to avoid unnecessary misunderstanding and 2 additional references to verify the mode of action of the ELISA kit. (line 548-556)
Round 3
Reviewer 2 Report (New Reviewer)
Comments and Suggestions for Authors
Although changing the title, still thoriught the manuscript "NK cell activity" is widely present, This must be changed. Thus, thorough text editing must be performed.
Also, it it very unusual to put p values into the abstract, please correct this.
Comments on the Quality of English LanguageThorough text edition is needed.
Author Response
Comments and Suggestions for Authors
Although changing the title, still thoriught the manuscript "NK cell activity" is widely present, This must be changed. Thus, thorough text editing must be performed.
Per your review, I have changed “NK cell activity” into IFN-γ concentration or secretion, and all of which are highlighted as blue ones.
Also, it it very unusual to put p values into the abstract, please correct this.
Even though those p values were inserted after other reviewer’s recommendation, I deleted all as you recommended.
Round 4
Reviewer 2 Report (New Reviewer)
Comments and Suggestions for Authors
I am satisfied with the corrections.
Comments on the Quality of English LanguageThe language is fine.
This manuscript is a resubmission of an earlier submission. The following is a list of the peer review reports and author responses from that submission.
Round 1
Reviewer 1 Report
Comments and Suggestions for Authors
The authors provide evidence for the peripheral natural killer cell activity and various invasive breast cancer clinical aspects. The authors have 396 samples, which provide significant value to the article. Some of the comments are:
1. The authors do not provide any explicit justification for the IFNg groups they chose to compare each other with. It will be necessary to provide some reasoning behind the choice. The tables also should be indicating < and >= instead of <=. Please check the table headers. What did the distribution of the IFNg values look like? H
2. Could the authors please provide a flowchart of how they arrived at 396 numbers after eliminating other patients? Line 83-84, are these 200 patients' part of the 337 patients?
3. Did the author test any Cox proportional hazard analysis with stage/outcome from treatment? How was the survival fraction calculated? The graphs are challenging to see. Also, these should be combined into one figure with panel and descriptive legends.
4. Please provide the reference of the formula for “BSA (body surface area) (m2) by DuBois formula" line 185
Author Response
- The authors do not provide any explicit justification for the IFNg groups they chose to compare each other with. It will be necessary to provide some reasoning behind the choice.
As described in Materials and Methods, this study used results from a commercial ELISA kit called NK Vue-Kit (lines 107-108), and the interpretation of the results was based on the results provided by the company that manufactures and sells the ELISA kit (NKMAX, line 108) (NK activity cut-off value at 100, 250, 500 pg/ml). (lines 144-146) As you pointed out, we contacted the manufacturer and asked for a reference with cut-offs at 100, 250, and 500 pg/ml, but were unable to obtain one. Therefore, we refer to reference 15 and present all results in four bins as well as two groups based on 100, 250, and 500 pg/ml. (line 156-160). I am thinking of next step analysis with these ideas using a lot more patients of various stages comparing IFN-γ values as continuous variable, which must be time-consuming effort to get the real cut-off value of IFN-γ for all stages and possibly in each stage. That must be totally different research.
- The tables also should be indicating < and >= instead of <=. Please check the table headers.
As you pointed out, I changed all <= into >=.
What did the distribution of the IFNg values look like? H
I don’t understand your question fully. As you know, the distribution pattern shall vary according to your specific condition to array the IFNγvalues. Regardless of complex clinical parameters, the percentage of each interval seems almost same, except exceptionally high percentage of more than 500pg/ml, roughly 4~5 times higher.
- Could the authors please provide a flowchart of how they arrived at 396 numbers after eliminating other patients? (Line 79~85)
Total of 1, 149 patients recruited.
753 patients were excluded:
337 patients missed NK value at diagnosis
278 patients treated with neoadjuvant chemotherapy
100 patients were ductal carcinoma-in-situ
38 patients with bilateral breast cancer
- 396 patients were entered for study. (1149-753=396, 337+278+100+38=753)
I think my article has already much tables with numerous figures. I don’t think it wise to add one another. But, if you insist, I will be happy to add consort diagram or flow chart.
Line 83-84, are these 200 patients' part of the 337 patients? No. 278 patients (more than 200) were excluded due to having neoadjuvant chemotherapy, and other 337 patients were also excluded for missing NK value at diagnosis.
- Did the author test any Cox proportional hazard analysis with stage/outcome from treatment?
Unfortunately, in this analysis, we failed to get meaningful relations with NK activity and survival, even after log-rank analysis (Mantel-Cox). I believe you are very familiar with statistical evaluations with much complex methods such as Cox proportional hazards model. If I got some possible link to go further statistical analysis of survival, I would do that as your recommendation. For me, this is the first study with NK ELISA kit to apply for breast cancer patients to predict poor prognosis better and possibly to select proper patient for immunotherapy with skyrocketing medical expenses. I am thinking of next step analysis with these ideas using a lot more patients of various stages comparing IFN-γ values as continuous variable, which must be time-consuming effort to get the real cut-off value of IFN-γ for all stages and possibly in each stage. That must be totally different research.
How was the survival fraction calculated? We used log-rank analysis (Mantel-Cox).
The graphs are challenging to see. Also, these should be combined into one figure with panel and descriptive legends.
I amended figure 1. As you requested.
- Please provide the reference of the formula for “BSA (body surface area) (m2) by DuBois formula" line 185
Du Bois D, Du Bois E. Clinical calorimetry: tenth paper a formula to estimate the approximate surface area if height and weight be known. Arch Intern Med 1916;17:863-71.

Reviewer 2 Report
Comments and Suggestions for Authors
Review comment
The manuscript titled "Peripheral Natural Killer Cell Activity in Invasive Breast Cancer at Diagnosis" tries to evaluate the correlation between peripheral NK cell activity and clinicopathological features in breast cancer patients. Authors assessed the correlation between the pretreatment peripheral blood NK cell activity measured by released interferon-gamma and the clinicopathological characteristics of patients diagnosed with curable invasive breast cancer. They also concluded that “As these treatments are increasingly likely to be used for breast cancer in the future, it is meaningful to study whether peripheral blood NK cell activity tests, which are simpler and easier to obtain than tumor-infiltrating NK cell analysis by pathologists, can be used objectively to predict prognosis and select patients for immunotherapy, once it is possible to determine a meaningful relationship between pretreatment NK cell activity and prognosis before starting full-scale treatment”. Certain novelty exists in this study. Authors collected plenty of data in this study, but apparent significance for clinical practice seems missing. Some major discussion is indispensable and needs to be added. Major concerns have been listed as following.
1.One of the major issues brought about by this article is “the efficiency of peripheral blood NK cell activity tests and tumor-infiltrating NK cell analysis by pathologists on prognosis prediction and patient selection for immunotherapy”. However, authors have not tested and compared the levels of “NK cell analysis by pathologist”. Thus, how to determine the advantages possessed by peripheral blood NK cell activity tests over NK cell analysis by pathologist on clinical outcomes prediction?
2.(DOI: 10.1016/S2095-4964(16)60275-3) is recommended to be cited right after "In that study even though showing clinical data from 39 patients they used 52Cr-labeling method to measure peripheral NK cell activity and presented possible role of systematically activated NK cells to improve cancer cell elimination".
3.Authors need to comprehensively and thoroughly discuss their conclusion “stage III was significantly related with lowest NK cell activity”. For instance, How the NK cell activity in stage I, II and even IV? Especially, why the lowest NK cell activity level has not been found in stage IV?
4. The limitations and challenges of applying peripheral blood NK cell activity tests on prognosis prediction and patient selection for immunotherapy should also be discussed based on current studies.
5.The potential mechanisms of how Peripheral Natural Killer Cell Activity affecting the prognosis of patients of Invasive Breast Cancer, should be elucidated by a figure.
6. The title “Peripheral natural killer cell activity in invasive breast cancer at diagnosis” should be refined and adjusted to the main conclusion of this manuscript.
7. The current epidemiological situation of breast cancer, along with its prognosis and treatment methods, needs to be summarized in section of introduction.
Author Response
- One of the major issues brought about by this article is “the efficiency of peripheral blood NK cell activity tests and tumor-infiltrating NK cell analysis by pathologists on prognosis prediction and patient selection for immunotherapy”. However, authors have not tested and compared the levels of “NK cell analysis by pathologist”. Thus, how to determine the advantages possessed by peripheral blood NK cell activity tests over NK cell analysis by pathologist on clinical outcomes prediction?
I completely agree with your points. The purpose of this study was not to replace or claim superiority over tumor-infiltrating NK cell analysis (TIL) by pathologists, so no comparison was made between the two. However, as you are well aware, there are quite a few papers on TIL that show diametrically opposed results, and I'm sure you would agree that what is and is not possible varies from institution to institution depending on the capabilities of the pathologist. Unfortunately, due to the limited number of breast cancer-certified pathologists at our institution, the study you recommend is not feasible, and instead the main objective was to find a correlation between established and uncontroversial clinical factors and peripheral NK cell activity.
2.(DOI: 10.1016/S2095-4964(16)60275-3) is recommended to be cited right after "In that study even though showing clinical data from 39 patients they used 52Cr-labeling method to measure peripheral NK cell activity and presented possible role of systematically activated NK cells to improve cancer cell elimination".
I found and read the above paper and it's about 'Effects of biofield therapy on a murine breast cancer model', but I inserted lines 291-297 as a reference to the statement that the ELISA kit used in this study is simpler and safer than the old method of using radioisotope. If I know more about what you are recommending, I would be happy to add it as a reference.
- Authors need to comprehensively and thoroughly discuss their conclusion “stage III was significantly related with lowest NK cell activity”. For instance, How the NK cell activity in stage I, II and even IV?
I agree with your point. It is true that IFN-γ <100 pg/ml in stage III was significantly different from the rest of the bands (100-250, 250-500, >500 pg/ml). The possible association with stages in Table 1. was further analyzed dividing NK activity into two bands, instead of using cut-off values of 100, 250, and 500 pg/ml (Table 2.), and finally confirmed by Bonferroni adjustment in Table 4. Naturally, stages I, II, and IV were also analyzed, and a statistically significant difference was found only in stage III.
Especially, why the lowest NK cell activity level has not been found in stage IV?
It would have been nice to be able to meaningfully see the same trends in stage IV as in stage III, but given that there were situations where the number of patients in stage IV at diagnosis was so small that the statistics were problematic, we would expect to see similar results if we could get a sufficient number of stage IV patients to allow for statistics.
- The limitations and challenges of applying peripheral blood NK cell activity tests on prognosis prediction and patient selection for immunotherapy should also be discussed based on current studies.
These results alone do not support the immediate use of peripheral NK cell activity test results in conjunction with established clinicopathologic factors to predict immunotherapy indications or prognosis in breast cancer. Re-establishing appropriate cut-off values in a larger number of diverse patients, or identifying precise cut-off values for each stage, would be a step forward. Research must continue on how to identify the optimal patients for costly immunotherapy in triple-negative breast cancer, which currently has the poorest prognosis and for which effective treatments are still in development, and in advanced breast cancer patients with recurrence and metastasis, for whom conventional treatments have reached their limits.
5.The potential mechanisms of how Peripheral Natural Killer Cell Activity affecting the prognosis of patients of Invasive Breast Cancer, should be elucidated by a figure.
This is a retrospective clinical study with patients’ data, and I believe that what you are recommending can only be done after a lot of data are available from in vitro studies and animal studies. However, I do know that there are a number of clinical trials of immunotherapy using natural killer cells that are being or will be registered on clinicaltrials.gov, so I expect that the exact and precise mechanisms of this will be disclosed in detail as the work of many great researchers pays off.
- The title “Peripheral natural killer cell activity in invasive breast cancer at diagnosis” should be refined and adjusted to the main conclusion of this manuscript.
What about “The lowest peripheral natural killer cell activity can be expected in stage III breast cancer patients” for the title of my research?
- The current epidemiological situation of breast cancer, along with its prognosis and treatment methods, needs to be summarized in section of introduction.
Per your recommendation, we're adding the following.
"Globally, breast cancer is the most common cancer threatening women's health, and over the past two decades, improvements in diagnostics and the development of taxanes and targeted agents have led to significant improvements in treatment, with a positive impact on survival rates. However, for triple-negative breast cancer, which has the poorest prognosis due to its lack of hormone receptors and poor response to targeted agents, recent clinical studies are slowly raising hopes for the potential of immunotherapy in addition to conventional treatment."

Round 2
Reviewer 2 Report
Comments and Suggestions for Authors
The authors failed to modify and reply to the previous revision suggestions, so that the level of evidence of this study has not been improved. Also, the clinical applications and significance hypothesized by this article still seem speculative and may not be robust, suggesting a reconsideration of its publication suitability.
Author Response
Dear reviewer,
First of all, I would like to say that I do appreciate your valuable review to make my manuscript more clear and worthy of reading. I have definitely sent you a file (ijms-2859073 (2).docx) with my response to your comments on February 3, 2024, and I suspect that the manuscript editor had problems delivering this file to you. I am resubmitting it with my best responses and corrections to your valuable points, and if you have any further corrections that you would like me to make, please let me know, and I will be happy to respond to them. Green highlights are for you and yellow ones for another reviewer.
My replies are the following below:
The manuscript titled "Peripheral Natural Killer Cell Activity in Invasive Breast Cancer at Diagnosis" tries to evaluate the correlation between peripheral NK cell activity and clinicopathological features in breast cancer patients. Authors assessed the correlation between the pretreatment peripheral blood NK cell activity measured by released interferon-gamma and the clinicopathological characteristics of patients diagnosed with curable invasive breast cancer. They also concluded that “As these treatments are increasingly likely to be used for breast cancer in the future, it is meaningful to study whether peripheral blood NK cell activity tests, which are simpler and easier to obtain than tumor-infiltrating NK cell analysis by pathologists, can be used objectively to predict prognosis and select patients for immunotherapy, once it is possible to determine a meaningful relationship between pretreatment NK cell activity and prognosis before starting full-scale treatment”. Certain novelty exists in this study. Authors collected plenty of data in this study, but apparent significance for clinical practice seems missing. Some major discussion is indispensable and needs to be added. Major concerns have been listed as following.
- One of the major issues brought about by this article is “the efficiency of peripheral blood NK cell activity tests and tumor-infiltrating NK cell analysis by pathologists on prognosis prediction and patient selection for immunotherapy”. However, authors have not tested and compared the levels of “NK cell analysis by pathologist”. Thus, how to determine the advantages possessed by peripheral blood NK cell activity tests over NK cell analysis by pathologist on clinical outcomes prediction?
I completely agree with your points. The purpose of this study was not to replace or claim superiority over tumor-infiltrating NK cell analysis (TIL) by pathologists, so no comparison was made between the two. However, as you are well aware, there are quite a few papers on TIL that show diametrically opposed results, and I'm sure you would agree that what is and is not possible varies from institution to institution depending on the capabilities of the pathologist. Unfortunately, due to the limited number of breast cancer-certified pathologists at our institution, the study you recommend is not feasible, and instead the main objective was to find a correlation between established and uncontroversial clinical factors and peripheral NK cell activity.
2.(DOI: 10.1016/S2095-4964(16)60275-3) is recommended to be cited right after "In that study even though showing clinical data from 39 patients they used 52Cr-labeling method to measure peripheral NK cell activity and presented possible role of systematically activated NK cells to improve cancer cell elimination".
I found and read the above paper and it's about 'Effects of biofield therapy on a murine breast cancer model', but I inserted lines 291-297 as a reference to the statement that the ELISA kit used in this study is simpler and safer than the old method of using radioisotope. If I know more about what you are recommending, I would be happy to add it as a reference.
- Authors need to comprehensively and thoroughly discuss their conclusion “stage III was significantly related with lowest NK cell activity”. For instance, How the NK cell activity in stage I, II and even IV?
I agree with your point. It is true that IFN-γ <100 pg/ml in stage III was significantly different from the rest of the bands (100-250, 250-500, >500 pg/ml). The possible association with stages in Table 1. was further analyzed dividing NK activity into two bands, instead of using cut-off values of 100, 250, and 500 pg/ml (Table 2.), and finally confirmed by Bonferroni adjustment in Table 4. Naturally, stages I, II, and IV were also analyzed, and a statistically significant difference was found only in stage III.
Especially, why the lowest NK cell activity level has not been found in stage IV?
It would have been nice to be able to meaningfully see the same trends in stage IV as in stage III, but given that there were situations where the number of patients in stage IV at diagnosis was so small that the statistics were problematic, we would expect to see similar results if we could get a sufficient number of stage IV patients to allow for statistics.
- The limitations and challenges of applying peripheral blood NK cell activity tests on prognosis prediction and patient selection for immunotherapy should also be discussed based on current studies.
These results alone do not support the immediate use of peripheral NK cell activity test results in conjunction with established clinicopathologic factors to predict immunotherapy indications or prognosis in breast cancer. Re-establishing appropriate cut-off values in a larger number of diverse patients, or identifying precise cut-off values for each stage, would be a step forward. Research must continue on how to identify the optimal patients for costly immunotherapy in triple-negative breast cancer, which currently has the poorest prognosis and for which effective treatments are still in development, and in advanced breast cancer patients with recurrence and metastasis, for whom conventional treatments have reached their limits.
5.The potential mechanisms of how Peripheral Natural Killer Cell Activity affecting the prognosis of patients of Invasive Breast Cancer, should be elucidated by a figure.
This is a retrospective clinical study with patients’ data, and I believe that what you are recommending can only be done after a lot of data are available from in vitro studies and animal studies. However, I do know that there are a number of clinical trials of immunotherapy using natural killer cells that are being or will be registered on clinicaltrials.gov, so I expect that the exact and precise mechanisms of this will be disclosed in detail as the work of many great researchers pays off.
- The title “Peripheral natural killer cell activity in invasive breast cancer at diagnosis” should be refined and adjusted to the main conclusion of this manuscript.
What about “The lowest peripheral natural killer cell activity can be expected in stage III breast cancer patients” for the title of my research?
- The current epidemiological situation of breast cancer, along with its prognosis and treatment methods, needs to be summarized in section of introduction.
Per your recommendation, we're adding the following.
"Globally, breast cancer is the most common cancer threatening women's health, and over the past two decades, improvements in diagnostics and the development of taxanes and targeted agents have led to significant improvements in treatment, with a positive impact on survival rates. However, for triple-negative breast cancer, which has the poorest prognosis due to its lack of hormone receptors and poor response to targeted agents, recent clinical studies are slowly raising hopes for the potential of immunotherapy in addition to conventional treatment."
Submission Date
21 January 2024
Date of this review
01 Feb 2024 10:10:24
